# The use of feasibility studies for stepped-wedge cluster randomised trials: protocol for a review of impact and scope

Caroline A Kristunas,[1] Karla Hemming,[2] Helen C Eborall,[1] Laura J Gray[1]

## ABSTRACT

**Introduction** The stepped-wedge cluster randomised trial (SW-CRT) is a complex design, for which many decisions about key design parameters must be made during the planning. These include the number of steps and the duration of time needed to embed the intervention. Feasibility studies are likely to be useful for informing these decisions and increasing the likelihood of the main trial's success. However, the number of feasibility studies being conducted for SW-CRTs is currently unknown. This review aims to establish the number of feasibility studies being conducted for SW-CRTs and determine which feasibility issues are commonly investigated.

**Methods and analysis** Fully published feasibility studies for SW-CRTs will be identified, according to predefined inclusion criteria, from searches conducted in Ovid MEDLINE, Scopus, Embase and PsycINFO. To also identify and gain information on unpublished feasibility studies the following will be contacted: authors of published SW-CRTs (identified from the most recent systematic reviews); contacts for registered SW-CRTs (identified from clinical trials registries); lead statisticians of UK registered clinical trials units and researchers known to work in the area of SW-CRTs. Data extraction will be conducted independently by two reviewers. For the fully published feasibility studies, data will be extracted on the study characteristics, the rationale for the study, the process for determining progression to a main trial, how the study informed the main trial and whether the main trial went ahead. The researchers involved in the unpublished feasibility studies will be contacted to elicit the same information. A narrative synthesis will be conducted and provided alongside a descriptive analysis of the study characteristics.

**Ethics and dissemination** This review does not require ethical approval, as no individual patient data will be used. The results of this review will be published in an open-access peer-reviewed journal.

## Strengths and limitations of this study

- ► This review will be the first to provide an insight into how feasibility studies are being used to inform stepped-wedge cluster randomised trials (SW-CRTs).
- ► This review will identify both published and unpublished feasibility studies for SW-CRTs.
- ► To ensure a robust review and minimise potential bias, the search strategy and inclusion and exclusion criteria have been prespecified.
- ► Although steps have been taken to minimise potential sources of bias, selection bias may be introduced through the exclusion of non-English language studies.

[1]Department of Health Sciences, University of Leicester, Leicester, UK
[2]Institute of Applied Health Research, University of Birmingham, Birmingham, UK

**Correspondence to**
Dr Laura J Gray; lg48@le.ac.uk

## INTRODUCTION
### The stepped-wedge cluster randomised trial

The stepped-wedge cluster randomised trial (SW-CRT) is seeing an unprecedented increase in its use.[1–4] For this design, clusters are randomised to start the intervention at different time points. All of the clusters start the trial not receiving the intervention, and then switch to the intervention at their allocated start time, until by the end of the trial all of the clusters are receiving the intervention figure 1.[5]

There are several appeals of the SW-CRT. One of the main appeals being that it allows a staggered introduction of the intervention, allowing clusters to act as their own control and for all clusters to eventually receive the intervention.[2 6] The introduction of the intervention is also randomised, making this design more desirable than a simple (non-randomised) before-and-after design, which are known to be confounded by temporal trends that cannot be adjusted for.[7] In addition, there are occasions when the SW-CRT is more efficient than a parallel CRT, requiring a smaller sample size and fewer clusters.[7 8] The SW-CRT can thus allow the experimental assessment of the effectiveness of certain interventions that, due to practical, logistical or financial reasons, it may not be possible to assess using another design of trial.[5 7 9] An example of a study for which the SW-CRT was considered the best option is the Sedation

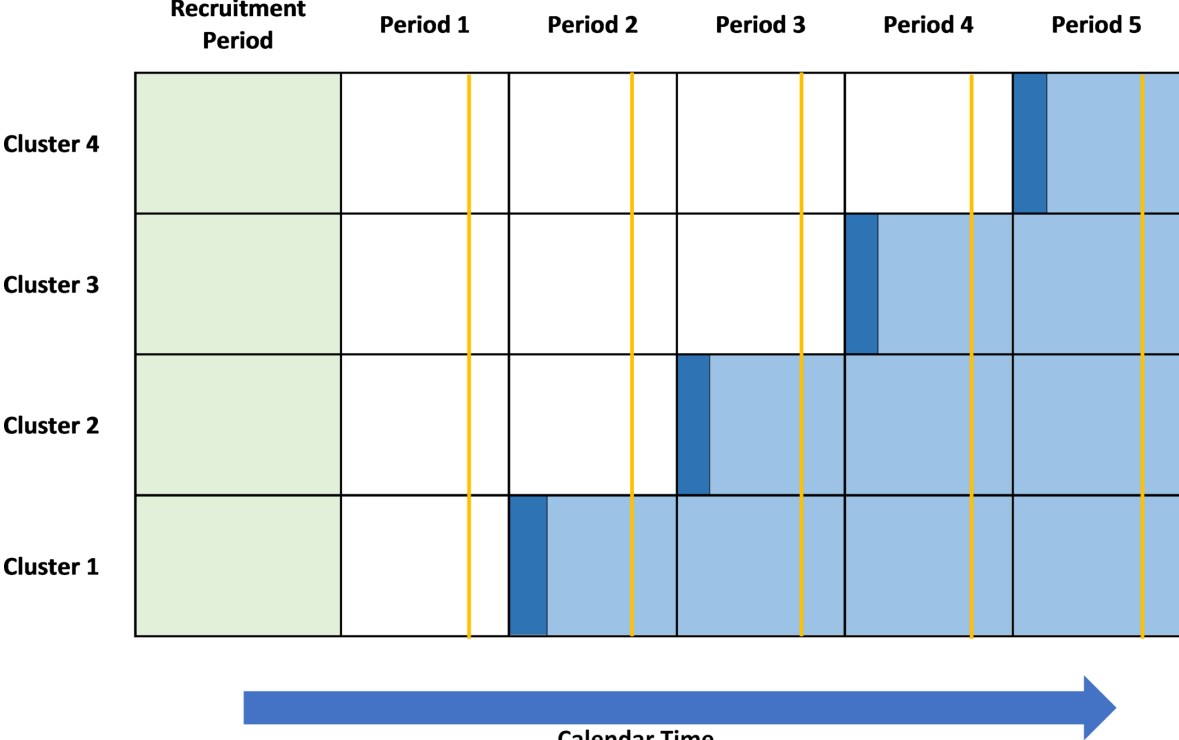

**Figure 1** Schematic of an example stepped-wedge cluster randomised trial (SW-CRT), with four steps and one cluster switching from control (white) to the intervention (light blue) at each time point. A recruitment period (light green) has been included prior to the first measurement period, as well as a bedding-in period (dark blue) during which the intervention will be embedded within the cluster. A cross-sectional sample is taken during each measurement period, at the time point indicated by the orange line, from which the outcome measure is obtained. Not all SW-CRTs follow the design shown. For example, some SW-CRTs may not include a recruitment or bedding-in period and others may be of a cohort design.

AND Weaning In CHildren (SANDWICH) trial.[10] Details of the reasons for this choice of design and the decision processes related to this choice are described in figure 2.

### Trial decision process in the SW-CRT

The staggered nature of the introduction of clusters to the intervention in a SW-CRT requires many decisions to be made during the design process, such as the number of start times and the length of time between start times.[7] For example there may be logistical constraints that mean it is only feasible to switch a certain number of clusters at once, or the intervention may take a while to embed in a cluster, in which case the length of time between start times might need to be sufficiently long to allow for this.[7] Alternatively, the trial might need to be completed within a fixed funding period, in which case the step length will need to be short enough to allow all clusters to switch within this period. These decisions are in addition to those that must be made for a standard CRT and are unlikely to be able to be informed by previous studies. It is known that, although advantageous in some respects, the staggered introduction of the intervention may increase the complexity of the trial in both practical and statistical terms.[6 7] Some of the issues associated with a staggered implementation are being investigated as part of a feasibility study for the SANDWICH trial, details of which are given in figure 2.

### What feasibility and pilot studies may offer

The terms pilot and feasibility study are often used interchangeably to describe a small scale study, that is conducted prior to the main trial and which aims to guide the planning or design of the trial or to confirm its feasibility.[11–14] When designing a trial, decisions may be informed by previous trials, systematic reviews, routine data and so on, but there may be aspects of the trial that require additional information which might only be obtained through the use of a feasibility or pilot study.[13] Pilot or feasibility studies are becoming the norm prior to a definitive individually randomised trial, and test trial factors such as recruitment, retention and acceptability of the intervention.[15 16]

Cluster randomised trials (CRTs) are typically larger and more expensive, and due to their additional complexity, tend to need more assumptions and decisions to be made at the design stage than for an individually randomised trial.[13 17] Each aspect of the design that is assessed at an individual level, now needs to also be assessed at the cluster level. These may include recruitment and retention rates or the acceptability of the intervention to both the individuals and the clusters. Specifically for CRTs, an estimate of the extent of variation between clusters (intracluster correlation coefficient (ICC)) is needed to calculate the required

sample size. However, Eldridge *et al*[17] have shown that feasibility studies to investigate ICCs for CRTs will often have insufficient accuracy, unless the feasibility study is almost as large as the main trial.

Feasibility studies may be of even greater potential benefit for SW-CRTs than they are for standard CRTs, as numerous additional decisions must be made. These include determining the number of clusters it is feasible to switch to the intervention at each step and the length of time between each step needed to embed the intervention, ensuring that the full effect of the intervention is realised and enough participants are recruited for each stage of the trial.[9]

Previous systematic reviews have investigated the reasons why a pilot or feasibility study might be undertaken for a randomised controlled trial (RCT),[11 15 18] but these reviews are unlikely to have captured many pilot or feasibility studies for SW-CRTs. In addition, no published systematic reviews of SW-CRTs have included feasibility studies, so it is not currently known to what extent feasibility studies are being conducted for SW-CRTs nor what issues these studies are being designed to investigate.

## What our review will achieve

We aim to identify what feasibility work is being conducted for SW-CRTs. This will be achieved through a systematic review of the published literature, which will identify published feasibility studies for SW-CRTs. However, as feasibility studies often go unpublished, we will extend our review to identify unpublished feasibility studies. This will be achieved by contacting the authors of published and registered SW-CRTs, lead statisticians of UK registered clinical trials units (CTUs) and researchers who are known by the authors to work in the area of SW-CRTs.

## OBJECTIVES

The overarching aims of this review are to determine the number of feasibility studies being conducted for SW-CRTs and how these studies are being used to inform the subsequent SW-CRTs. Specifically, our objectives are to:

1. Systematically identify published feasibility studies for SW-CRTs;

---

### Case study: the SANDWICH trial

The **S**edation **AND W**eaning **I**n **CH**ildren (SANDWICH) trial, will be a cluster-randomised stepped-wedge clinical and cost-effectiveness trial set in UK Paediatric Intensive Care Units (PICU). A protocol-based intervention, incorporating co-ordinated care with greater nursing involvement in managing sedation and weaning ventilation, will be assessed for its effectiveness at reducing the duration of invasive mechanical ventilation.

There are several reasons why a stepped-wedge cluster randomised design was chosen for the SANDWICH trial. Firstly, it was important that randomisation was undertaken at the cluster level, as the intervention will be delivered at the PICU (cluster) level and contamination would have occurred if individual-level randomisation had been used. Secondly, it would have been infeasible to conduct a conventional parallel cluster trial. There are a maximum of 20 PICUs in the UK, limiting the number of available clusters to such an extent that there would not have been a sufficient number to detect the important clinical effect at the desired 90% power.

It is anticipated that, due to the staggered rollout of the intervention, there will be particular issues that will need to be overcome. It is desired to conceal the allocation sequence, as advance knowledge of the allocation sequence can result in clusters withdrawing if they are allocated to switch to the intervention late in the trial. A feasibility study, taking the form of an internal pilot, will be used to determine how close the crossover date can be revealed to each cluster, to allow them sufficient time to prepare. This study will also aim to confirm the feasibility of data collection procedures and delivery of the intervention; adherence to the randomisation sequence by the PICUs; fidelity of uptake and acceptability of the intervention and determine whether any tweaks to the intervention are required.

**Figure 2** A case study of the SANDWICH stepped-wedge cluster randomised trial.

2. Identify unpublished feasibility work that has been conducted or is planned to take place prior to a SW-CRT;
3. Extract information on the design characteristics and rationale for these feasibility studies, as well as how these studies inform the main trial and how the decision is made as to whether to progress to the main trial.

## METHODS

A two-pronged approach will be implemented to identify feasibility studies for SW-CRTs. Published feasibility studies will be identified through a systematic review of the literature, whereas unpublished feasibility studies will be identified by contacting researchers who may have been involved in, or have knowledge of, feasibility studies for SW-CRTs that have been conducted but which have not yet been published.

### Search strategy

#### Identification of published feasibility studies for SW-CRTs

We will identify eligible feasibility studies for SW-CRTs, published in English, via electronic searches of the online published databases Ovid MEDLINE (1946–present), Scopus (1966–present), Embase (1947–present) and PsycINFO (1967–present). The initial search strategy will include a combination of MeSH terms and keywords specific to each bibliographic database. An example search strategy is outlined in Box 1 and is based on previously published search strategies.[1–3 11 15]

---

**Box    1 Example search strategy for Ovid MEDLINE**

1. 'pilot*'.mp
2. 'feasibil*'.mp
3. 1 OR 2
4. 'step* wedge*'.mp
5. 'step*wedge*'.mp
6. 'delay* intervention'.mp
7. 'experimental* staged introduction'.mp
8. ('one* direction* crossover design' OR 'one* direction* cross* over design').mp
9. ('incremental* recruitment' OR 'incremental* introduction' OR 'incremental* implementation' OR 'incremental* allocation').mp
10. ('phased* recruitment' OR 'phased* introduction' OR 'phased* implementation' OR 'phased* allocation').mp
11. ('staggered* recruitment' OR 'staggered* introduction' OR 'staggered* implementation' OR 'staggered*allocation').mp
12. ('stepwise* recruitment' OR 'stepwise* introduction' OR 'stepwise* implementation' OR 'stepwise*allocation').mp
13. ('step*wise* recruitment' OR 'step*wise* introduction' OR 'step*wise* implementation' OR 'step*wise*allocation').mp
14. ('delayed* recruitment' OR 'delayed* introduction' OR 'delayed* implementation' OR 'delayed*allocation').mp
15. or/4–14
16. 3 AND 15
17. limit 16 to English language

---

The titles and abstracts for the identified studies will be screened for eligibility by two reviewers independently in a random order, with full-text articles being obtained for potentially eligible studies and the same duplicate method of assessment used. Any studies which are not found to be eligible will be excluded and the reason for exclusion noted. Any differences of opinion will be resolved by a third reviewer if required. Authors will be contacted if any further information is required and we will aim to access any published or unpublished protocols for each of the identified feasibility studies.

Reference lists of the identified studies will also be checked for potentially eligible studies. Any additional published feasibility studies for SW-CRTs which are known to the authors, but which are not identified during the database searches, will also be included and assessed for eligibility. Records will be managed using Refworks reference management software.

### Identification of unpublished feasibility studies for SW-CRTs

Feasibility studies for SW-CRTs might not always be published. In an attempt to identify the majority of feasibility studies for SW-CRTs, our systematic review will be extended to allow inclusion of unpublished feasibility studies. This will be achieved in the following ways:

► Unpublished feasibility work for published SW-CRTs will be identified from the SW-CRTs reported in the most recent systematic reviews. The trial reports and references will be examined for evidence of feasibility work. The authors of these trials will then be contacted to identify whether any unreported feasibility work was conducted.
► Feasibility work that has been conducted for SW-CRTs that are yet to be published will be sought by contacting the lead statisticians of CTUs, researchers working in the area of stepped-wedge trials, as well as through searches of clinical trials registries.

### Inclusion and exclusion criteria

For the systematic review of published feasibility studies, eligible studies will be full reports or protocols of feasibility studies for SW-CRTs published during the period covered by the searched databases (1946–present). For the unpublished feasibility studies, no publications will be required to qualify for inclusion.

The definition of, and distinction between feasibility and pilot studies is not always clear, with the terms often being used interchangeably[11 12] and conflicting definitions existing for the distinction between the terms.[14 19] For the purpose of this review (including both published and unpublished studies), a feasibility study will be defined as a study, with clearly defined aims and objectives, which intends to ascertain the feasibility of a planned SW-CRT, through the assessment of issues other than solely the effectiveness and refinement of the intervention, such as investigating recruitment issues. Since pilot studies will also often investigate issues affecting the feasibility of the main trial, they will be considered to be a type of

feasibility study for the purpose of this review, provided they satisfy the criteria of a feasibility study.

The feasibility study itself does not need to be of a stepped-wedge design, or even randomised, as the design will depend on the objectives of the study. However, the study should, through focused objectives, make it clear how the findings of the feasibility study will inform the main study, which must be intended to be of an SW-CRT design. An SW-CRT will be defined as a randomised trial, which randomises clusters and which has two or more steps (time-points at which clusters change treatment group). Studies where the planned definitive trial is individually randomised, has a bidirectional cross-over design or is non-randomised, will be excluded. Neither the published nor the unpublished feasibility studies will be limited to healthcare settings.

The researchers involved in the unpublished feasibility studies that we identify will be contacted to gain the information required to determine eligibility.

## DATA EXTRACTION
Data for eligible published feasibility studies will be extracted by two reviewers independently and in random order using a data extraction form that will have been tested on a small number of studies, before being refined and finalised. Any differences will be resolved though discussion with a third reviewer. Extracted data will be managed in Microsoft Excel V.2013.

For the eligible unpublished feasibility studies, data will be obtained through semistructured interviews with the researchers involved in the studies. This will be conducted by a single reviewer. The same data will be collected for the unpublished feasibility studies as is extracted for the published feasibility studies.

### Data extraction of trial characteristics
Data will be extracted on aspects of the design of the feasibility study, including the recruitment, randomisation, blinding and overall design (parallel, stepped-wedge, etc), as well as how exposure to the intervention is experienced at an individual level. For example, in some studies subjects might receive both the control and intervention conditions, whereas in others they might only experience the one condition. Data will also be extracted on how each feasibility study defines itself (either as a pilot or feasibility study), the size of the feasibility study and how the chosen sample size is justified.

### Data extraction of study rationale
Data will be extracted on the rationale for conducting a feasibility study prior to the main trial. This will be achieved by extracting the specific aims of each study and categorising them into process, resource, management and scientific motivations.[20] These categories will be defined as:
- ► Process—assesses the feasibility of the steps that need to take place during the main study. This will include the estimation of recruitment, consent and retention rates, testing of the process and acceptability of the randomisation procedure, testing of the exclusion criteria, identification of barriers to recruitment, adherence and so on. Some of these will be specific to the SW-CRT, for example, the recruitment rates per measurement period, which will determine the length of time between steps, whereas others will be more generic.
- ► Resource—assesses the time and budget problems that may occur during the main trial. This may include the time taken for data collection for example, time taken to mail or fill out forms/surveys and have them returned or the time taken for data entry to be completed. The human resources available may also be a consideration, ensuring that the right people are available with the required expertise. Specific to SW-CRT, these aims might include the length of time required to roll out the intervention or the time taken to acquire the necessary permissions from each of the clusters.
- ► Management—assesses the potential for human and data optimisation problems, such as the feasibility of international collaborations and across site coordination for multicentre trials.
- ► Scientific motivations—assesses the scientific processes such as treatment safety and acceptability and estimation of the ICC (not recommended[17]), potential effectiveness and variance parameters.

In addition, the aims of each feasibility study will be subdivided into aims that are specific to the SW-CRT and those which will be common with other trial designs.

### Data extraction of progression to main trial
Data will be extracted on the process for deciding whether the study will progress to the main trial. This will include the type of analysis conducted, either hypothesis testing or descriptive, how much emphasis is put on the results of any hypothesis tests and the criteria used for determining success (main trial is considered feasible). Information on any hard or soft stopping rules that are in place for the feasibility studies will also be extracted.

### Data extraction of main trial characteristics
Data will be extracted on the details of the main trials that follow each of the feasibility studies; whether it goes ahead, how the feasibility study has informed or made changes to the main trial and whether any of the participants who took part in the feasibility study also took part in the main trial.

## ANALYSIS OF RESULTS
We will present a narrative synthesis of our findings, as well as a descriptive analysis of the study characteristics of each of the eligible feasibility studies that we identify for inclusion in the review.

## ETHICS AND DISSEMINATION

As no individual patient data will be used, this review does not require ethical approval. We intend on presenting the findings of this review in an open-access peer-reviewed publication in an appropriate journal. We also intend on disseminating the findings through presentations at relevant conferences.

## DISCUSSION

Although several reviews have been conducted in the area of stepped-wedge trials,[1–4 21 22] none have investigated the use of feasibility studies for SW-CRTs. Due to their complexity, there are likely to be many issues that will affect the feasibility of SW-CRTs, some of which may be common across studies. However, it has yet to be reported how often feasibility work is being conducted prior to SW-CRTs, nor what feasibility issues these studies have been designed to inform. We aim to use our review to determine how many feasibility studies are conducted for SW-CRTs and what issues these studies have been designed to investigate. From the number of feasibility studies that we identify, we will be able to infer the rough order of magnitude of SW-CRTs which have some form of feasibility work conducted prior to the main trial. This will provide an estimate of how often feasibility studies are conducted prior to an SW-CRT.

By prespecifying our search strategy, the inclusion and exclusion criteria and ensuring that two reviewers independently review all studies in a randomised order, we aim for our review to be robust, minimising potential sources of bias. Since the term 'stepped wedge' has not been universally used to describe a trial of this design, we have included in our search strategy other terms used to describe this design, terms which have been used in previous reviews.[1–4 22] However, there still remains the potential for selection bias in our review, as only trials using one of these terms will be included, which will exclude any SW-CRTs using alternative terminology. We may also introduce selection bias through the exclusion of non-English language studies, although there is evidence that this has a minimal effect.[23]

Our review would be limited if relying solely on published feasibility studies for SW-CRTs, as many will not have been published, particularly if proceeding to the main trial was regarded as infeasible. However, by extending our systematic review to identify unpublished feasibility studies for SW-CRTs, we will be able to widen the reach of our review. We will contact authors of published SW-CRTs and researchers known to be working in the area of SW-CRTs to determine if they have knowledge of additional studies that we can include in our review. It may be anticipated that there will be few feasibility studies for SW-CRTs. However, with the various methods we plan on implementing, we expect to be able to identify the majority of those feasibility studies for SW-CRTs that do exist. We have already identified a handful of these studies from a small scoping review and some studies that

we already knew about. We are therefore optimistic that our full review will be able to identify further studies.

## CONCLUSION

This review is the first in a series of related projects investigating the feasibility of SW-CRTs. It will determine the number of feasibility studies being conducted to inform SW-CRTs and identify the feasibility issues that are being investigated and how these studies are informing the main SW-CRTs. This will provide an insight into how feasibility studies can benefit SW-CRTs. Future work will identify the feasibility issues being encountered in SW-CRTs and ultimately lead to the development of guidance on how feasibility studies for SW-CRTs can be conducted.

**Contributors** CAK conceptualised the review, the design of which was refined by incorporating comments from KH, HCE and LG. CAK drafted the manuscript and incorporated comments from KH, HCE and LG for all subsequent drafts. All authors approved the final version and agreed to be accountable for all aspects of the work.

**Funding** This report is independent research supported by the National Institute for Health Research (NIHR Doctoral Research Fellowship, Miss Caroline Kristunas, DRF-2016-09-025). The authors would also like to acknowledge support from the NIHR Collaboration for Leadership in Applied Health Research and Care—East Midlands (NIHR CLAHRC–EM), NIHR CLAHRC–West Midlands, the Leicester Clinical Trials Unit and the NIHR Leicester-Loughborough Diet, Lifestyle and Physical Activity Biomedical Research Unit, which is a partnership between University Hospitals of Leicester NHS Trust, Loughborough University and the University of Leicester. The views expressed are those of the authors and not necessarily those of the NHS, the National Institute for Health Research or the Department of Health.

**Competing interests** None declared.

**Provenance and peer review** Not commissioned; externally peer reviewed.

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
