## [Reviewer comments · BMJ Open]

ARTICLE DETAILS

TITLE (PROVISIONAL)	The use of feasibility studies for stepped-wedge cluster randomised trials: a protocol for a review of impact and scope.
AUTHORS	Kristunas, Caroline; Hemming, Karla; Eborall, Helen; Gray, Laura

VERSION 1 - REVIEW

REVIEWER	Mike Campbell University of Sheffield UK I am currently on a CONSORT working group with Karla Hemming which we hope will result in a paper.
REVIEW RETURNED	16-May-2017

GENERAL COMMENTS	This study overlaps two of my main interests so although I may not be conflicted, I may be well be biased! Since the piece is a protocol, there is not much to criticise. I would not be surprised if the authors failed to find any relevant studies (or a mere handful of poorly described ones) and they do not discuss this possibility. I wondered if they may consider an initial search on feasibility studies for cluster trials and then drill down to stepped wedge if they find any in the first sweep.
--

REVIEWER	Richard Parker Senior Statistician, University of Edinburgh, United Kingdom
REVIEW RETURNED	30-May-2017

GENERAL COMMENTS	1. One of the overarching aims of the review is “to determine how often feasibility studies are conducted for SW-CRTs” (lines 140 and 303) but I am not clear how this aim would be fully accomplished without getting a handle on how many SW-CRTs did NOT perform a feasibility study prior to the main trial. It is true that you will be able to get a good estimate of the *number* of feasibility studies conducted, but the question of “how often” is a step up from this and I interpret this as meaning you are interested in “what proportion” of SW-CRTs have conducted feasibility studies prior to the main trial? Am I correct, and if so, please clarify how this objective would be achieved.2. One of the main appeals of the SW-CRT design is not just that it allows a “staggered introduction of the intervention” (line 76) but that it allows a *randomised* introduction of the intervention: i.e. it allows the intervention to be introduced to the clusters at random time points. Otherwise, there is not much advantage of using the design over a simple before-and-after study. I think this should be
--

	emphasized in the introduction section. 3. Figure 1 shows a “typical” schematic of a SW-CRT, but not all SW-CRT are in exactly the same format as that shown in Figure 1. Please clarify this in the figure legend. 4. Regarding the inclusion/exclusion criteria (lines 190-213), I think these could be stated a little clearer. How many years will you search back to? Will any studies be excluded because they are too old (e.g. before the methodology was fully developed)? Also in lines 199 to 200, the implication is that any studies assessing “the effectiveness and refinement of the intervention” will be excluded. But how about those studies which are “assessing the effectiveness and refinement of the intervention” as only one of many different objectives. Will such studies still be included in the review? 5. Regarding the data extraction of trial characteristics (lines 224 to 228), will you also extract data on the type of SW-CRT since there are many different types? For example, some SW-CRT include subjects which receive both the control and intervention conditions, whereas others will have subjects which experience only one intervention/control condition depending on the way subjects are recruited. You might also consider stratifying the analysis by the type of SW-CRT. Minor points: 6. Abstract line 47. I find the phrase “data will be extracted on the general characteristics” quite vague on first reading. General characteristics of what? 7. Line 95. Typo: “is” should be “in”. 8. Lines 178-180. “In an attempt to identify the anticipated majority of these studies,...” The precise meaning is not clear. 9. Line 289. “...as many will not have been published, particularly if the main trial was found to be infeasible”. This sentence is semantically difficult because there are two possible interpretations. I think you mean “proceeding to a planned main trial was regarded as infeasible”, but the phrase could alternatively be interpreted as meaning that the main trial has already been conducted and found to be infeasible and so therefore the feasibility study was not published.
--	---

VERSION 1 – AUTHOR RESPONSE

Responses to Reviewer 1's comments.

Comment:

This study overlaps two of my main interests so although I may not be conflicted, I may be well be biased!

Since the piece is a protocol, there is not much to criticise.

I would not be surprised if the authors failed to find any relevant studies (or a mere handful of poorly described ones) and they do not discuss this possibility. I wondered if they may consider an initial search on feasibility studies for cluster trials and then drill down to stepped wedge if they find any in the first sweep.

Thank you for your kind and helpful comments, they have led to an improved manuscript. We are glad to hear you share our interest in this area, but we understand your concern. However, prior to the writing of the protocol for this review, we were aware of the FRIDOM-feasibility study, DECIDE-LVAD pilot study, an internal pilot for the SANDWICH trial and the piloting of the FOXY intervention, all of which precede a planned stepped-wedge cluster randomised trial. In addition, a scoping review also identified the SOCLE II pilot study. Although we do not expect to find a large number of relevant studies we think that the range of methods that we plan to use will identify the majority of the studies that do exist. In the manuscript, this point is now discussed on lines 309-314.

Responses to Reviewer 2's comments.

Thank you for your constructive comments and suggestions, they have enabled us to improve the manuscript. We have addressed each of these in turn below.

1. One of the overarching aims of the review is “to determine how often feasibility studies are conducted for SW-CRTs” (lines 140 and 303) but I am not clear how this aim would be fully accomplished without getting a handle on how many SW-CRTs did NOT perform a feasibility study prior to the main trial. It is true that you will be able to get a good estimate of the *number* of feasibility studies conducted, but the question of “how often” is a step up from this and I interpret this as meaning you are interested in “what proportion” of SW-CRTs have conducted feasibility studies prior to the main trial? Am I correct, and if so, please clarify how this objective would be achieved.

We apologise for the misleading wording. It is true that our review will aim to determine the number of feasibility studies being conducted for SW-CRTs and not the proportion as is suggested. The wording throughout the manuscript has now been changed to reflect this. However, whilst we will not be able to determine exactly what proportion of SW-CRTs conduct some sort of feasibility study, knowledge of the absolute number of SW-CRTs conducted (from other reviews) will allow us to infer the rough order of magnitude of SW-CRTs which have some sort of feasibility work conducted before the main trial. We have mentioned this in the discussion section (lines 292-295).

2. One of the main appeals of the SW-CRT design is not just that it allows a “staggered introduction of the intervention” (line 76) but that it allows a *randomised* introduction of the intervention: i.e. it allows the intervention to be introduced to the clusters at random time points. Otherwise, there is not much advantage of using the design over a simple before-and-after study. I think this should be emphasized in the introduction section.

Thank you for highlighting this important point to us. This has now been added to the introduction section (lines 77-80).

3. Figure 1 shows a “typical” schematic of a SW-CRT, but not all SW-CRT are in exactly the same format as that shown in Figure 1. Please clarify this in the figure legend.

Thank you for bringing to our attention that the original figure legend was misleading. It has now been clarified in the legend for Figure 1 that the SW-CRT shown is only an example and that other SW-CRTs may take a different format.

4. Regarding the inclusion/exclusion criteria (lines 190-213), I think these could be a stated a little clearer. How many years will you search back to? Will any studies be excluded because they are too old (e.g. before the methodology was fully developed)? Also in lines 199 to 200, the implication is that any studies assessing “the effectiveness and refinement of the intervention” will be excluded. But how

about those studies which are “assessing the effectiveness and refinement of the intervention” as only one of many different objectives. Will such studies still be included in the review?

Thank you for highlighting these oversights to us. All studies identified from the search of the databases will be included. Ovid MEDLINE goes back the furthest, to 1946 and so we will include studies published from 1946 to the time of the search. This detail has been added to the manuscript in lines 195-196.

We intended for studies to be excluded if their sole objective was to assess the effectiveness and refinement of the intervention. We therefore intend to include those studies assessing the effectiveness and refinement of the intervention in addition to other objectives that relate to the feasibility of the study. We have clarified this in the manuscript (lines 203-204).

5. Regarding the data extraction of trial characteristics (lines 224 to 228), will you also extract data on the type of SW-CRT since there are many different types? For example, some SW-CRT include subjects which receive both the control and intervention conditions, whereas others will have subjects which experience only one intervention/control condition depending on the way subjects are recruited. You might also consider stratifying the analysis by the type of SW-CRT.

Thank you. We do indeed intend on extracting data on the type of SW-CRT and this has now been added to the manuscript (lines 231-234).

Minor points:

6. Abstract line 47. I find the phrase “data will be extracted on the general characteristics” quite vague on first reading. General characteristics of what?

Thank you for bringing this to our attention. In order to clarify that information on the general characteristics of the feasibility study will be extracted this phrase has been changed to “data will be extracted on the general study characteristics” (line 47).

7. Line 95. Typo: “is” should be “in”.
Typo has been corrected (line 97), thank you.

8. Lines 178-180. “In an attempt to identify the anticipated majority of these studies,…” The precise meaning is not clear.

We agree that this sentence is unclear and have changed to “in an attempt to identify the majority of feasibility studies for SW-CRTs…” (line 181).

9. Line 289. “...as many will not have been published, particularly if the main trial was found to be infeasible”. This sentence is semantically difficult because there are two possible interpretations. I think you mean “proceeding to a planned main trial was regarded as infeasible”, but the phrase could alternatively be interpreted as meaning that the main trial has already been conducted and found to be infeasible and so therefore the feasibility study was not published.

Thank you for highlighting this issue. This sentence has now been altered to remove the possible misinterpretation. It now reads: “...as many will not have been published, particularly if proceeding to the main trial was regarded as infeasible.” (lines 304-305).

We wish to thank you and the reviewers again for your time considering our manuscript and for the useful comments and suggestion, that we hope you will agree have led to an improved manuscript.